# AeroLight: A Lightweight Architecture with Dynamic Feature Fusion for High-Fidelity Small-Target Detection in Aerial Imagery

**DOI:** 10.3390/s25175369

**Published:** 2025-08-30

**Authors:** Hao Qiu, Xiaoyan Meng, Yunjie Zhao, Liang Yu, Shuai Yin

**Affiliations:** 1School of Computer and Information Engineering, Xinjiang Agricultural University, Urumqi 830052, China; 320243429@stu.xjau.edu.cn (H.Q.); zyj@xjau.edu.cn (Y.Z.); 320243435@stu.xjau.edu.cn (L.Y.); 320243432@stu.xjau.edu.cn (S.Y.); 2Ministry of Education, Engineering Research Center for Intelligent Agriculture, Urumqi 830052, China; 3Xinjiang Agricultural Informatization Engineering Technology Research Center, Urumqi 830052, China

**Keywords:** small-target detection, UAV aerial images, AeroLight, dynamic feature fusion, lightweight

## Abstract

Small-target detection in Unmanned Aerial Vehicle (UAV) aerial images remains a significant and unresolved challenge in aerial image analysis, hampered by low target resolution, dense object clustering, and complex, cluttered backgrounds. In order to cope with these problems, we present AeroLight, a novel and efficient detection architecture that achieves high-fidelity performance in resource-constrained environments. AeroLight is built upon three key innovations. First, we have optimized the feature pyramid at the architectural level by integrating a high-resolution head specifically designed for minute object detection. This design enhances sensitivity to fine-grained spatial details while streamlining redundant and computationally expensive network layers. Second, a Dynamic Feature Fusion (DFF) module is proposed to adaptively recalibrate and merge multi-scale feature maps, mitigating information loss during integration and strengthening object representation across diverse scales. Finally, we enhance the localization precision of irregular-shaped objects by refining bounding box regression using a Shape-IoU loss function. AeroLight is shown to improve mAP50 and mAP50-95 by 7.5% and 3.3%, respectively, on the VisDrone2019 dataset, while reducing the parameter count by 28.8% when compared with the baseline model. Further validation on the RSOD dataset and Huaxing Farm Drone dataset confirms its superior performance and generalization capabilities. AeroLight provides a powerful and efficient solution for real-world UAV applications, setting a new standard for lightweight, high-precision object recognition in aerial imaging scenarios.

## 1. Introduction

The proliferation of Unmanned Aerial Vehicles (UAVs) has revolutionized data acquisition across numerous domains, enabling high-resolution earth observation for applications in precision agriculture [1,2,3], intelligent transportation systems [4,5], disaster response [6,7], and infrastructure monitoring [8,9,10]. Object detection constitutes a fundamental component in the analysis of large-scale aerial imagery and underpins critical applications such as environmental monitoring and security surveillance. However, despite recent advancements, the reliable detection of small-scale targets continues to pose substantial challenges. In high-altitude imagery, objects including vehicles, pedestrians, and minor infrastructure elements often occupy only a limited number of pixels, thereby increasing their susceptibility to feature degradation and information loss. This problem is exacerbated by dense object clustering, complex background clutter, and drastic variations in scale and orientation, which collectively degrade the performance of conventional object detection methods [11,12].

In recent years, deep learning-based detectors, particularly one-stage architectures like the YOLO series [13,14,15], have become the de-facto standard due to their exceptional balance of speed and performance. However, these general-purpose models frequently underperform when applied to aerial imagery. Their conventional architectures, originally optimized for ground-level visual data, employ aggressive downsampling of feature maps, which inevitably leads to the loss of critical fine-grained spatial details essential for accurate small-object recognition.

Furthermore, their feature pyramid networks (FPNs) [16] employ simple fusion strategies that can be insufficient to harmonize features across the vast scale disparities inherent in aerial viewpoints.

Consequently, a substantial body of research has emerged to adapt these architectures to UAV applications. Efforts have largely focused on three fronts: enhancing multi-scale representation by adding more detection layers [17] or specialized fusion modules [18]; creating lightweight models suitable for on-board deployment using techniques like efficient convolutions [19] or network pruning [20]; and incorporating attention mechanisms to focus on salient target regions [21,22]. Despite these incremental advances, most existing approaches tend to address the aforementioned challenges in isolation. A holistic solution that simultaneously enhances small-target sensitivity, optimizes feature fusion, and maintains a lightweight profile for real-time, on-board deployment remains an open problem. The field still lacks a dedicated framework that systematically co-designs these components for the unique constraints of aerial vision.

To mitigate these concerns, we created AeroLight, a novel architecture that systematically re-engineers the detection pipeline for the specific demands of aerial vision. Our approach begins with a fundamental redesign of the feature hierarchy. We replace the conventional, computationally expensive low-resolution detection head (P5) with a new, high-resolution head (P2) augmented with an attention mechanism. This strategic reallocation of the computational budget prioritizes the preservation of fine-grained spatial details, enabling the model to capture minute targets that are typically lost during aggressive downsampling. To complement this structural enhancement and address information degradation during feature fusion, we propose a novel Dynamic Feature Fusion (DFF) module. Integrated into the model’s neck, the DFF module employs a parallel dual-attention mechanism that adaptively recalibrates feature maps along both spatial and channel dimensions. Through adaptive emphasis on salient object traits and suppression of noisy backgrounds, the model produces more reliable and distinguishable multi-scale features. Finally, to refine the model’s localization capabilities, we replace the standard CIoU loss with a more advanced Shape-IoU regression objective. By explicitly accounting for the geometric properties and scale consistency of bounding boxes, this loss function provides more accurate supervision, significantly improving localization precision for the irregularly shaped and oriented targets frequently encountered in aerial imagery. These innovations synergistically create a highly efficient and accurate detector tailored for real-world, resource-constrained UAV applications.

## 2. Related Work

We position our research at the junction of general object detection, small-target detection in aerial scenes, lightweight model design, and advanced feature fusion techniques. This section surveys recent advances in these fields to better position AeroLight’s contributions.

### 2.1. General Object Detection Architectures

Modern object detection has been dominated by deep learning, which is broadly divided into two paradigms. Two-stage detectors, such as R-CNN [23] and its successor Faster R-CNN [24] first generate candidate regions, then classify and refine each. Despite high accuracy, their multi-step process brings significant computational overhead, making them impractical for time-sensitive applications. In contrast, one-stage detectors perform prediction in a single pass, directly regressing object bounding coordinates and category confidence values inferred from raw input image data. This family includes the Single Shot Detector (SSD) [25] and the You Only Look Once (YOLO) series [13,14,15]. YOLO, in particular, has emerged as the dominant benchmark for real-time detection, owing to its exceptional balance between speed and accuracy. More recently, transformer-based models like DETR [26] have emerged, casting detection as a bipartite matching problem between sets and removing dependency on hand-tuned components such as non-maximum suppression (NMS). Prior to this, RetinaNet [27] introduced the Focal Loss function, which down-weights the contribution of a large number of easy negative samples in standard cross-entropy, enabling the model to focus more on hard-to-classify samples during training and thus effectively improving detection accuracy. However, these foundational models are primarily designed for general-purpose datasets where objects are well-defined and of medium-to-large scale, and their performance degrades significantly in the challenging domain of aerial imagery.

### 2.2. Small-Target Detection in Aerial Imagery

The detection of small targets in UAV imagery is a long-standing challenge due to low pixel counts, background clutter, and scale variance. To tackle this issue, researchers have investigated multiple strategies.

A primary approach involves enhancing multi-scale feature representation. The Feature Pyramid Network (FPN) [16] was a seminal work, but its top-down information flow is often insufficient for small objects. Consequently, many studies have focused on augmenting the feature pyramid. Some works, like YOLO-Drone [17], introduce additional, higher-resolution detection heads to capture finer details. Others design more complex bi-directional fusion paths, such as PANet [28] or BiFPN [29], to enrich feature semantics. Another line of research uses data-level enhancement. Methods like DSAA-YOLO [30] employ super-resolution techniques to artificially increase the resolution of small targets before feeding them to the detector. While effective, this adds pre-processing overhead. AeroLight builds on the architectural approach, but instead of simply adding a new head, we strategically reallocate the model’s capacity by simultaneously adding a P2 head and pruning the P5 head, creating a framework inherently biased towards small-object sensitivity without increasing complexity.

### 2.3. Lightweight and Efficient Detection Models

The deployment of detection models on resource-constrained UAV platforms necessitates lightweight architectures. A dominant strategy is the design of efficient convolutional blocks. Models like MobileNet [31], ShuffleNet [32], and GhostNet [33] have introduced depth-wise separable convolutions or channel splitting to reduce parameters and FLOPs. GCL-YOLO [19], for instance, leverages GhostConv to build a lightweight detector for UAVs, while RFAG-YOLO [11], despite achieving a balance between detection accuracy and model parameters, still leaves room for improvement. Other techniques include network pruning, quantization, and knowledge distillation, which compress a large, pre-trained model into a smaller one. While these methods are effective for general model compression, they are often task-agnostic. Our work contributes a more targeted approach to lightweight design. By analyzing the specific requirements of aerial detection (dominance of small targets), we achieve efficiency not through generic block replacement but through a deliberate and functional redesign of the feature pyramid, demonstrating that task-aware architectural choices can yield superior performance-efficiency trade-offs.

### 2.4. Feature Fusion and Attention Mechanisms

The optimal integration of hierarchical features is critical for merging fine-grained spatial patterns with abstract semantic representations. As mentioned, FPN and its variants established the foundation. However, the simple concatenation or summation used in these methods can lead to feature aliasing and information dilution. Attention mechanisms formulate this as a soft feature selection problem, where relevance scores determine the weights of the feature contribution. Channel attention, exemplified by Squeeze-and-Excitation (SE) Networks, learns to re-weight feature channels. Spatial attention centers on pinpointing the locations of critical features. The Convolutional Block Attention Module (CBAM) [21] serially combines both, refining features first along the channel dimension and then the spatial dimension, while LGFF-YOLO [34] enhances small object detection in UAV imagery via local–global feature fusion. AE-YOLO [35] performs feature fusion by selecting three-layer feature maps from the backbone network and employing a two-way weighted feature pyramid, where end-to-end network training learns the weight information of each branch to improve feature utilization. However, both the sequential processing in CBAM and such feature fusion strategies in LGFF-YOLO and AE-YOLO can be suboptimal. To address the demands of specific fusion scenarios, the latest specialized approaches have been proposed in recent studies: Hu et al. [36] proposed a complementarity-aware feature fusion method for aircraft detection, which generates pixel-aligned SAR images from unpaired optical data via SFEG, amplifies cross-modal differential features using DFSC, and dynamically fuses optical-SAR features with GWF; Zhu et al. [37] proposed the FDA-IRSTD framework, which extracts local frequency features via PFFT, enhances intra-frequency, inter-frequency and cross-window features using FAM, and dynamically fuses frequency-spatial features with FSFM. These methods highlight the value of task-specific fusion designs tailored to distinct data characteristics. Our proposed Dynamic Feature Fusion (DFF) module advances this concept by computing channel and spatial attention in parallel. The dual-path design facilitates a more straightforward and complementary feature recalibration process, mitigating the risk of information loss between steps. By integrating this module into the fusion neck, AeroLight achieves a more discriminative and noise-resistant feature representation, which plays a vital role in separating diminutive targets from intricate background interference in aerial imagery. Unlike previous works that append attention blocks, our DFF is designed as a core, integrative component of the dynamic process of feature fusion itself.

## 3. The AeroLight Architecture

Addressing the non-trivial task of small object identification in drone imagery, we propose AeroLight, a lightweight and high-fidelity detection architecture. AeroLight represents a fundamental architectural innovation rather than a simple refinement, with its core components completely reengineered to maximize both accuracy and computational efficiency in resource-limited scenarios. The architecture is built on three synergistic innovations: (1) a task-aware feature pyramid optimized for small objects, (2) a novel Dynamic Feature Fusion (DFF) module for robust multi-scale feature integration, and (3) a shape-aware regression loss for precise target localization. The overall architecture of AeroLight is depicted in Figure 1.

### 3.1. Task-Aware Feature Pyramid Redesign

Motivation: Standard object detectors, including the baseline YOLOv12, employ a feature pyramid with output heads at multiple scales (e.g., P3, P4, P5). However, in aerial scenes, small targets are far more prevalent than large ones. The P5 head, which processes heavily downsampled feature maps (e.g., 20 × 20), not only incurs significant computational costs but also suffers from a severe loss of the high-resolution structural information fundamental to miniature target recognition. This makes it largely redundant and inefficient for the target application.

Our Approach: We fundamentally reallocate the model’s computational budget by redesigning the feature pyramid to be inherently sensitive to small targets. As illustrated in Figure 2, our approach is two-fold:High-Resolution Head Addition: We introduce a new detection head at the P2 level (160 × 160 resolution), which operates on feature maps with a much smaller stride. This head is responsible for detecting the smallest objects, leveraging high-resolution features from the early stages of the backbone to preserve fine-grained details.Redundant Head Pruning: We completely remove the P5 large-target detection head. By employing this pruning strategy, model parameters and computational costs (FLOPs) are reduced considerably, without sacrificing detection performance for primary small-target groups.

To integrate the new P2 head, the feature map from the neck is upscaled by a factor of 2× via nearest-neighbor interpolation and concatenated with the corresponding P2 feature map from the backbone. This merged feature map then undergoes processing via an attention-based fusion block (A2C2f) before being passed to the detection head. This task-aware redesign creates a more efficient architecture that is explicitly optimized for the small-object detection paradigm typical of UAV imagery, attaining an exceptional equilibrium between model accuracy and resource efficiency.

### 3.2. Dynamic Feature Fusion (DFF) Module

Motivation: Multi-scale feature aggregation in FPN architectures proves particularly essential for remote sensing image analysis, but standard methods relying on simple concatenation and convolution can be suboptimal. They often fail to effectively suppress background noise and can dilute informative features, especially when fusing disparate scales.

Our Approach: We propose a novel Dynamic Feature Fusion (DFF) module designed to substitute the standard C3k2 blocks in the model’s neck. The DFF module, shown in Figure 3, employs a parallel, dual-attention mechanism to adaptively recalibrate feature maps along both spatial and channel dimensions. This allows the network to learn *what* to emphasize (channels) and *where* to focus (spatial locations) simultaneously.

The DFF module operates as follows: given two input feature maps $F_{1},F_{2}\in R^{C\times H\times W}$ from different scales, they are first processed to produce an intermediate feature map $F_{in}$. The module then splits into two parallel attention branches:Channel Attention Branch: This branch computes a channel attention vector $W_{ch}\in R^{C\times 1\times 1}$. The network employs spatial pooling at the global scale to aggregate contextual features, followed by a small multi-layer perceptron (MLP) and a sigmoid activation. This vector learns to selectively boost informative feature channels while suppressing less useful ones.Spatial Attention Branch: This branch computes a spatial attention map $W_{sp}\in R^{1\times H\times W}$. The network processes input features through convolutional operations, followed by a sigmoid activation, to generate a map that highlights salient spatial regions corresponding to potential targets.

Finally, the input feature map $F_{in}$ is recalibrated using the channel attention vector and then subjected to element-wise multiplication with the spatial attention map, resulting in the fused output $\hat{F}$. This parallel design ensures that spatial and channel recalibrations are complementary and do not interfere with each other, yielding enhanced feature discriminability essential for small target detection in cluttered environments.

### 3.3. Shape-Aware Bounding Box Regression

Motivation: The standard CIoU loss, while effective, primarily considers aspect ratio and center point distance. In aerial imagery, targets often appear with significant rotational and shape variations. The CIoU loss can struggle in these scenarios, leading to imprecise bounding box localization.

Our Approach: The CIoU loss is replaced by us with the more advanced Shape-IoU loss [38]. This loss function provides a more comprehensive geometric evaluation by explicitly considering the alignment of bounding box shapes, in addition to overlap and distance. The overall Shape-IoU loss, $L_{\mathrm{Shape}-\mathrm{IoU}}$, is formulated as:(1)$$L_{\mathrm{Shape}-\mathrm{IoU}}=1-\mathrm{IoU}\left(A,B\right)+\frac{d_{\mathrm{shape}}+\Omega_{\mathrm{shape}}}{2}$$
where *A* and *B* are the predicted and ground-truth boxes, $d_{\mathrm{shape}}$ is a distance penalty based on the box centers, and $\Omega_{\mathrm{shape}}$ is a shape-matching penalty. The shape penalty is calculated as:(2)$$\Omega_{\mathrm{shape}}=\sum_{t\in \{w,h\}}{\left(1-e^{-w_{t}}\right)}^{\theta }$$
where $w_{t}$ represents the discrepancy in width and height between the boxes, and θ is a hyperparameter controlling the focus on shape deviation. By decomposing the distance loss into directional components and adding a dedicated shape-matching term, Shape-IoU provides more stable and accurate gradients for regression. This leads to significantly improved localization accuracy, particularly for the small and irregularly oriented targets prevalent in aerial views.

## 4. Experimental Setup

The experimental configuration implemented to systematically validate AeroLight’s efficacy is presented herein. We describe the datasets used to assess accuracy and generalization, the implementation details for reproducibility, and the metrics for evaluating both the accuracy of object detection and the efficiency in computational operations.

### 4.1. Datasets

For a comprehensive evaluation, we utilize three distinct datasets, each addressing a wide range of challenges in aerial imaging. A selection of these datasets is presented in Figure 4.

VisDrone2019 [39]: This is our primary benchmark for evaluating small-target detection in complex urban environments. Captured by UAVs at various altitudes, it comprises 6471 training images, 548 validation images, and 1610 test images, spanning 10 object categories. Its defining characteristics are extremely dense scenes, significant scale variation, and severe object occlusion, making it an ideal testbed for the core challenges AeroLight aims to solve.

RSOD [40]: For generalization assessment, we conduct experiments on the RSOD benchmark dataset. It contains 976 images with 6,950 object instances across four distinct categories (aircraft, overpass, amusement park, fuel tank). The diverse object types and backgrounds test the model’s robustness beyond the urban scenes of VisDrone.

Huaxing Farm Drone Dataset: For additional generalization verification, we employ a custom-collected, domain-specific dataset comprising 876 aerial farmland samples, which specifically includes three object categories: cars, trucks, and pedestrians. These images were captured under clear weather conditions using a DJI Mavic 3M drone (DJI, Shenzhen, China) equipped with DJI Pilot 2 flight control software (Version 10.1.0.30). The flight altitude for this image collection was set at 110 m, ensuring the aerial images achieve a ground resolution better than 5 cm. Additionally, the collection parameters were configured with an 80% forward overlap and 70% side overlap. These images were captured from a UAV flying at an altitude exceeding 100 m, resulting in extremely small targets (e.g., cars, pedestrians) with very low pixel counts. This dataset specifically evaluates the model’s performance under challenging, high-altitude surveillance scenarios.

### 4.2. Implementation Details

All models were trained from scratch to ensure a fair comparison. The experiments were conducted using the PyTorch v2.2.2 framework on a server equipped with an NVIDIA GeForce RTX 4090 GPU (Santa Clara, CA, USA). For datasets such as VisDrone2019, RSOD, and our Huaxing Farm Drone Dataset, input images were adjusted to the fixed resolution of 640 × 640 using a letterbox resizing strategy (preserving original aspect ratios via black-border padding on the shorter side), with label coordinates scaled proportionally and offset by padding values to align with the 640 × 640 dimensions; during inference, predicted boxes were transformed back to original sizes by reversing these operations. Key hyperparameters used for training are summarized in Table 1. The experimental setup maintained uniform augmentation procedures—specifically mosaic tiling, mixup synthesis, and color spectrum adjustments—except where explicitly indicated.

### 4.3. Evaluation Metrics

Two core benchmarks were used to assess our model: detection accuracy and model efficiency, both of which are essential for practical UAV application.

Detection Accuracy. We use the standard mean Average Precision (mAP) as the primary metric.

mAP50: Mean average precision computed at the IoU = 0.5 threshold, measuring fundamental object detection performance.mAP50–95: Following the COCO evaluation protocol, this metric represents the mean average precision computed across 10 evenly-spaced IoU thresholds ranging from 0.5 to 0.95 (increments of 0.05). Provides a more comprehensive measure of localization accuracy, rewarding models that produce highly precise bounding boxes.Precision (P) and Recall (R) are also reported for a more detailed diagnostic analysis and to generate Precision-Recall (P-R) curves.

Model Efficiency. To quantify the suitability of our model for resource-constrained platforms, we measure:Parameters (Params): The total number of trainable parameters in the model, reported in millions (M). This metric reflects the model’s size and memory footprint.GFLOPs: Giga Floating-Point Operations per Second. This measures the computational complexity of a single forward pass, evaluated on a 640 × 640 input image. It is a key indicator of inference speed.

## 5. Results and Analysis

This section systematically evaluates AeroLight’s detection performance through comparative analysis with state-of-the-art methods on the VisDrone2019 benchmark. Subsequently, an in-depth ablation study is performed to analyze the contributions of each architectural improvement. Lastly, qualitative analyses and results from supplementary datasets are presented to highlight robustness and generalizability of the model.

### 5.1. Comparison with State-of-the-Art Methods

We evaluated AeroLight against a wide range of established and recent object detectors on the VisDrone2019 validation set. As shown in Table 2, AeroLight sets a new state-of-the-art for lightweight models in this challenging domain.

Our model achieves an mAP50 of 39.8% and an mAP50-95 of 22.6%, representing substantial gains of +7.5% and +3.3% respectively over the YOLOv12n baseline. Critically, these accuracy improvements are achieved while simultaneously reducing the parameter count by 28.8%. Compared to other high-performing lightweight models like YOLO11s, AeroLight delivers superior accuracy across all metrics with only 19.5% of the parameters and 54.0% of the GFLOPs. This demonstrates the effectiveness of our task-aware design philosophy, which prioritizes architectural intelligence over brute force. The consistent outperformance shown in the mAP50 training curve (Figure 5) further highlights AeroLight’s superior learning efficiency and convergence.

### 5.2. Ablation Study

For validating the effectiveness of each proposed component, an ablation study was systematically conducted on the VisDrone2019 dataset: our contributions were incrementally added to the YOLOv12n baseline. The outcomes, as outlined in Table 3, confirm that each module offers unique and synergistic benefits.

Pyramid Redesign (P2/P5): Optimal performance improvement is achieved through the dual strategy of P2 head integration coupled with P5 head removal, increasing mAP50 by +5.6% and mAP50–95 by +2.7%. This confirms our hypothesis that reallocating computational resources to high-resolution features is paramount for small-target detection, while also dramatically reducing the model size.Dynamic Feature Fusion (DFF): Integrating the DFF module alone improves mAP50 by +1.9%. This highlights its ability to create more discriminative feature representations by effectively fusing multi-scale information and suppressing background noise.Shape-IoU Loss: Adopting the Shape-IoU loss function yields a +1.6% gain in mAP50, underscoring its superior ability to handle the irregular shapes and orientations of aerial targets, thereby improving localization precision.

When all three components are combined, they work synergistically, resulting in the full AeroLight model which achieves a total gain of +7.5% in mAP50. This demonstrates that our contributions are complementary and collectively form a highly effective detection framework.

### 5.3. Qualitative Analysis

To provide an intuitive understanding of AeroLight’s advantages, we present qualitative comparisons.

Attention Visualization. The class activation maps (CAM) in Figure 6 reveal where the model focuses its attention. Compared to the baseline, AeroLight produces much sharper and more accurately localized activation maps that tightly envelop small and densely packed targets. This is a direct result of the DFF module’s ability to enhance salient features while filtering out irrelevant background clutter.

Detection Results. Figure 7 showcases detection outputs in four challenging scenarios from VisDrone2019. AeroLight consistently demonstrates superior performance by: (1) detecting more densely overlapping small objects; (2) detecting small, distant vehicles missed by the baseline and RT-DETR; (3) correctly distinguishing between similar classes (e.g., not misclassifying flags as pedestrians); and (4) maintaining robustness in adverse conditions like low light and object blur. These visual results corroborate the quantitative improvements, showing a clear reduction in both false negatives and false positives.

Confusion Matrix Cross-Model Comparison. Quantitative assessment of the confusion matrices presented in Figure 8 reveals distinct performance characteristics between the AeroLight and YOLOv12n models.Within these matrices, the ordinate axis corresponds to predicted class assignments, while the abscissa represents ground truth annotations.The principal diagonal elements, serving as the critical performance indicators, quantify the per-class true positive rate, thereby directly reflecting each model’s classification accuracy for individual object categories.Comparative analysis demonstrates that AeroLight systematically outperforms the YOLOv12n baseline across the majority of object classes.This consistent performance differential provides empirical evidence of AeroLight’s enhanced recognition capability, yielding superior overall detection and classification accuracy across diverse object types.

Per-Category Performance. An analysis of per-category accuracy (Figure 9) reveals that AeroLight’s improvements are most pronounced for the smallest object classes, such as pedestrians, motorcycles, and bicycles, where the mAP50 gain exceeds 10%. This disproportionate improvement confirms the effectiveness of our design in addressing the core challenge of low-pixel-ratio object detection.

### 5.4. Generalization Analysis

To verify that AeroLight’s capabilities are not confined to a single dataset, we evaluated its generalization performance on two additional, diverse datasets.

RSOD Dataset. On the RSOD dataset, which features different object types and remote sensing contexts, AeroLight again outperforms all baselines. As shown in Table 4, it achieves an mAP50 of 95.6%, demonstrating its ability to adapt to varied target appearances and backgrounds. The enhanced Precision-Recall curve in Figure 10 further validates its superior balance of precision and recall, a critical attribute for reliable real-world deployment.

Huaxing Farm Drone Dataset. This custom dataset represents an extreme test case with exceptionally small targets. The results in Table 5 show that AeroLight achieves an mAP50 of 83.7%, a significant +3.1% improvement over the baseline. The visual results in Figure 11 confirm this, with AeroLight successfully detecting vehicles that are nearly indistinguishable to the human eye, while the baseline fails. This strong performance under challenging, high-altitude conditions underscores the practical value and robustness of our architecture for real-world surveillance and monitoring tasks.

## 6. Discussion

In the task of detecting small objects in UAV aerial photography, there is a widespread trade-off between model parameters (Params) and accuracy (mAP). To achieve an efficient balance between the two, we improved the latest state-of-the-art (SOTA) object detection model and proposed the AeroLight algorithm. It enhances detection precision while achieving model lightweighting, laying a solid foundation for its deployment on edge devices. Meanwhile, the AeroLight algorithm has certain limitations: currently, it only supports RGB input and has not incorporated infrared or multispectral data, which have been widely adopted in recent UAV detection research. Another limitation is that the discussion on detection in extreme scenarios in the model design is still insufficient—although the current evaluation is based on standard datasets under typical conditions, future work will conduct rigorous assessments of the model’s robustness through specialized benchmarks such as datasets for object detection in foggy and cloudy scenarios [5]. This aims to quantify the performance degradation caused by weather factors. Future research plans to explore multimodal input fusion schemes and conduct in-depth analysis of the impact of weather factors.

## 7. Conclusions

In this work, we addressed the critical and persistent challenge of small-target detection in UAV aerial imagery by proposing AeroLight, a novel, lightweight, and high-fidelity detection architecture. Our approach holistically redesigns the standard detection pipeline through a synergistic combination of three key innovations: a task-aware feature pyramid that reallocates computational resources to high-resolution features for small-object sensitivity; a novel Dynamic Feature Fusion (DFF) module that ensures robust multi-scale integration by adaptively recalibrating spatial and channel information; and a Shape-IoU loss function that refines localization precision for irregularly shaped targets. Exhaustive experiments on the challenging VisDrone2019, RSOD, and Huaxing Farm Drone Dataset demonstrated that AeroLight outperforms existing state-of-the-art (SOTA) methods in overall performance. The proposed approach yields an enhancement of 7.5% mAP50 over baseline while reducing the parameters by 28.8%, demonstrating optimal precision-efficiency trade-offs. Consequently, AeroLight not only establishes a new, powerful benchmark for aerial detection, but also provides a practical and efficient solution ready for implementation on computationally limited UAV systems. This research paves the way for future advancements in on-board AI, where techniques such as automated network pruning and knowledge distillation can be explored to create the next generation of ultra-lightweight models, enabling fully autonomous, low-power perception for critical applications in smart agriculture, public safety, and infrastructure monitoring.

## Figures and Tables

**Figure 1 sensors-25-05369-f001:**
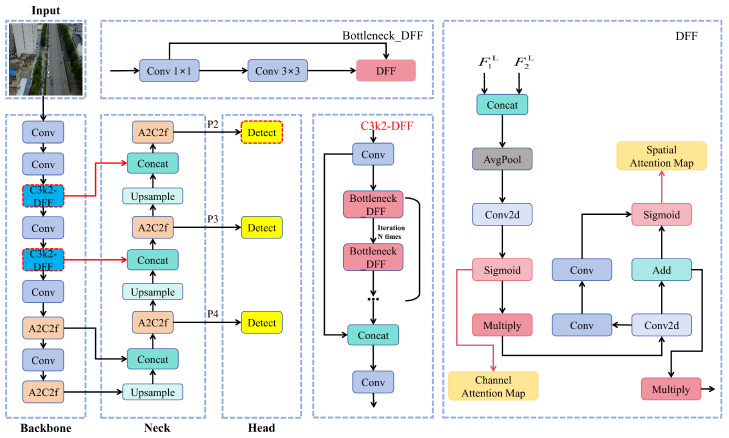
The overall architecture of the AeroLight model. The framework features a redesigned neck with a high-resolution P2 detection head (yellow) and prunes the P5 head. The core fusion logic is driven by our proposed Dynamic Feature Fusion (DFF) module (dark blue), which replaces standard convolutional blocks. The improved modules, including the high-resolution P2 detection head and the DFF module, are highlighted with red boxes for clear visualization.

**Figure 2 sensors-25-05369-f002:**
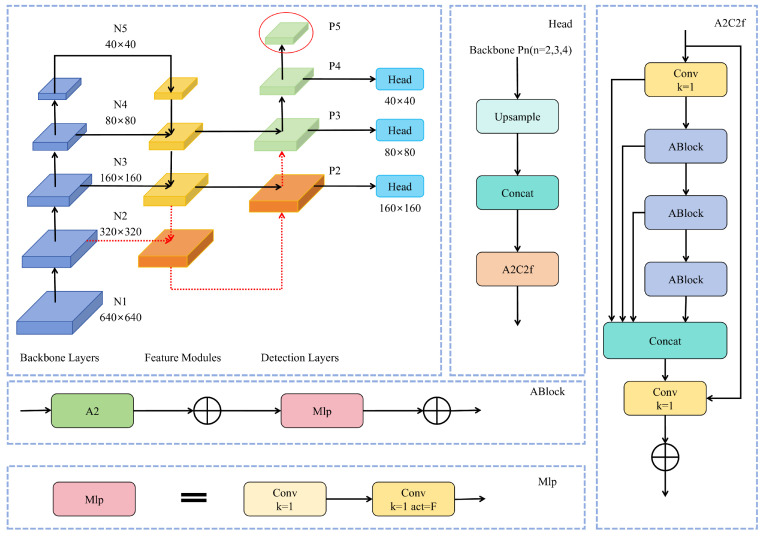
Schematic of the feature pyramid redesign. We introduce a new P2 detection head (indicated by red arrows) to enhance small-target sensitivity. Concurrently, the original P5 head is removed (indicated by the red ellipse) to reduce computational overhead. This creates a more efficient pyramid structure tailored for aerial scenes.

**Figure 3 sensors-25-05369-f003:**
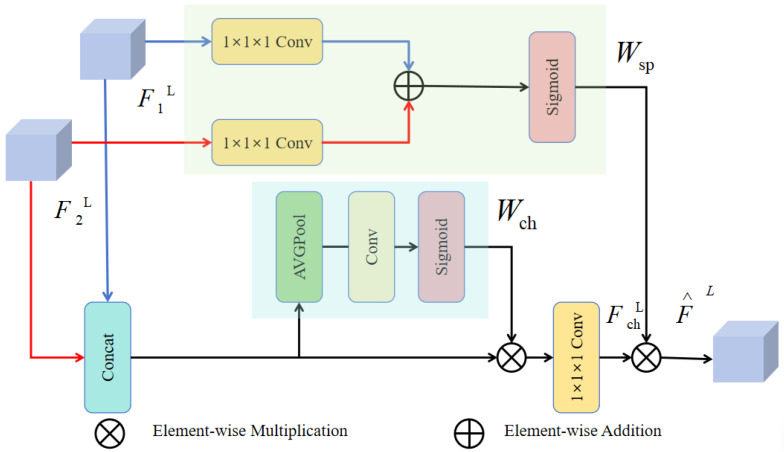
The architecture of the Dynamic Feature Fusion (DFF) module. It processes input features through parallel channel and spatial attention branches. The resulting attention weights ($W_{\mathrm{ch}}$ and $W_{\mathrm{sp}}$) are combined to adaptively refine the feature map, enhancing target representation while suppressing background clutter.

**Figure 4 sensors-25-05369-f004:**
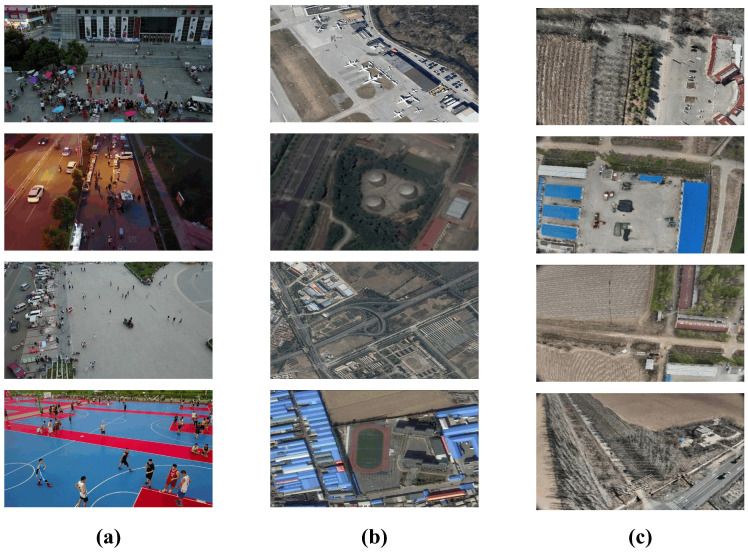
Sample images from the benchmark datasets. (**a**) VisDrone2019 features dense, multi-class scenes. (**b**) RSOD contains varied object categories in different remote sensing contexts. (**c**) Huaxing Farm Drone Dataset captures authentic aerial scenarios with diverse topographical features. All three illustrate key challenges like small targets and complex backgrounds.

**Figure 5 sensors-25-05369-f005:**
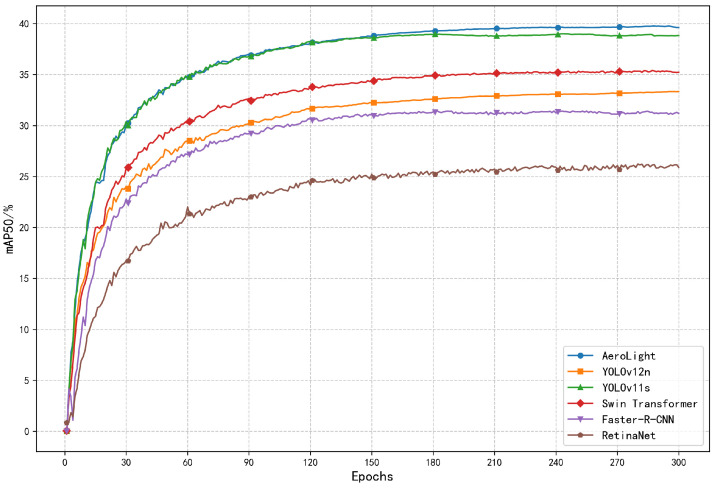
Training curves comparing the mAP50 of AeroLight against other leading models on VisDrone2019. AeroLight consistently achieves higher accuracy throughout the training process.

**Figure 6 sensors-25-05369-f006:**
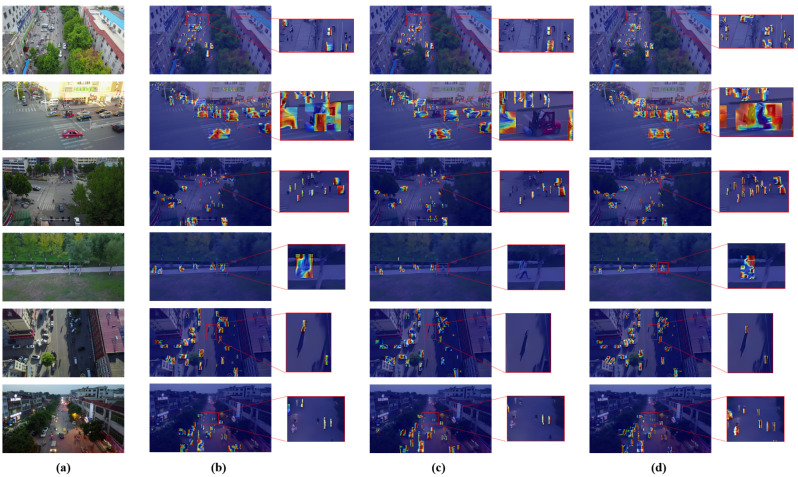
Heatmap comparison across different models. Our model generates more focused and precise attention maps, especially for small and clustered objects. (**a**) Original lmage. (**b**) RT-DETR. (**c**) YOLOv12n baseline. (**d**) AeroLight.

**Figure 7 sensors-25-05369-f007:**
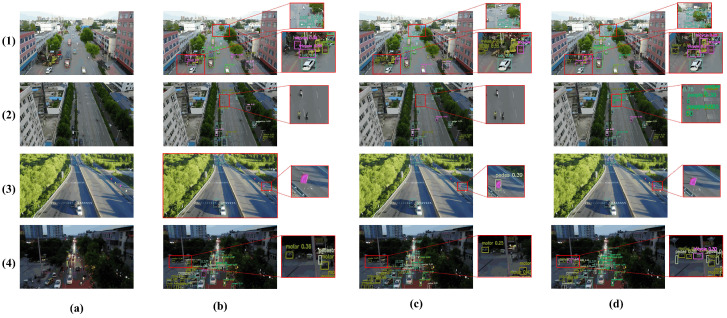
Visual comparison of detection performance across complex scenes from VisDrone2019. AeroLight successfully detects more small targets and reduces false positives compared to the baseline and RT-DETR. (**a**) Original lmage. (**b**) RT-DETR. (**c**) YOLOv12n baseline. (**d**) AeroLight.

**Figure 8 sensors-25-05369-f008:**
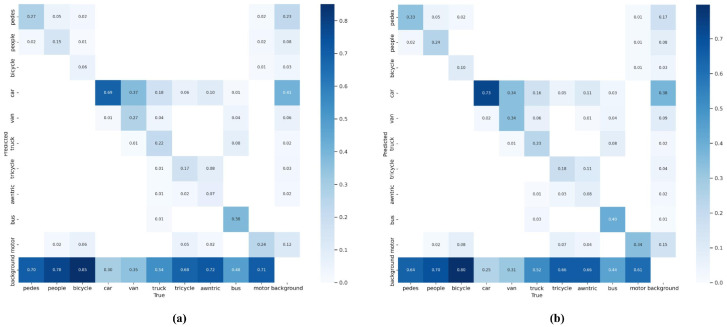
Confusion Matrix Comparison. AeroLight demonstrates stronger diagonal dominance in small-object classes (e.g., people, motor) compared to YOLOv12n baseline. (**a**) YOLOv12n baseline. (**b**) AeroLight.

**Figure 9 sensors-25-05369-f009:**
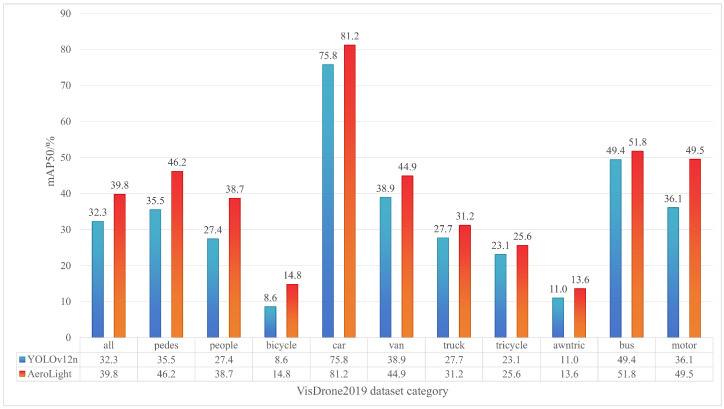
Per-category mAP50 comparison on VisDrone2019. AeroLight shows the most significant gains on small-object categories like 'pedestrian', 'motor', and 'bicycle'.

**Figure 10 sensors-25-05369-f010:**
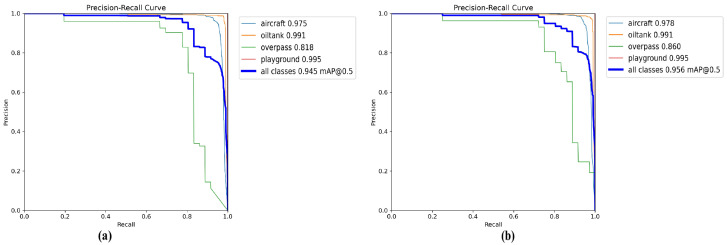
Precision-Recall curve comparison on the RSOD dataset. The larger area under the curve for AeroLight demonstrates a superior balance between detection precision and recall performance. (**a**) YOLOv12n baseline. (**b**) AeroLight.

**Figure 11 sensors-25-05369-f011:**
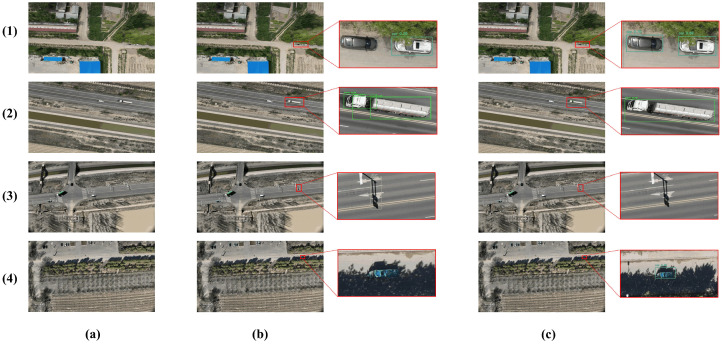
Detection results on the Huaxing Farm Drone Dataset. AeroLight outperforms the YOLOv12n baseline with higher detection accuracy, particularly in identifying extremely small vehicles that the baseline frequently misses, while also achieving a lower false detection rate. The enlarged views (right) highlight the detection quality. (**a**) Original lmage. (**b**) YOLOv12n baseline. (**c**) AeroLight.

**Table 1 sensors-25-05369-t001:** Key hyperparameters used for training all models.

Hyperparameter	Value
Optimizer	AdamW
Base Learning Rate	1 × 10^−3^
Weight Decay	0.05
Batch Size	16
Number of Epochs	300
Input Resolution	640 × 640

**Table 2 sensors-25-05369-t002:** Performance comparison with state-of-the-art models on the VisDrone2019 validation set. AeroLight achieves the best balance of accuracy and efficiency. Best values for lightweight models (Params < 10 M) are in bold.

Method	mAP50 (%)	mAP50–95 (%)	Params (M)	GFLOPs
RetinaNet [27]	26.3	12.9	21.4	101.3
Faster-R-CNN [24]	30.9	13.1	63.2	37.0
TOOD	31.9	19.7	32.4	112.3
YOLOv5s	32.5	18.7	7.1	23.8
YOLOv8n	32.1	19.4	3.01	8.1
RTMDet [41]	32.1	18.6	4.9	204.2
YOLOv10n	32.2	18.8	2.26	6.5
GFL [42]	32.2	18.5	33.3	/
YOLOv12n (Baseline)	32.3	19.3	2.57	6.5
Cascade R-CNN [43]	34.5	19.4	32.4	34.3
Swin Transformer	35.6	20.6	34.2	44.5
YOLO-Drone [17]	37.0	21.3	5.9	11.4
RT-DETR	37.7	21.8	19.9	57.0
LGFF-YOLO [34]	38.3	22.8	4.2	12.4
YOLO11s	38.7	22.2	9.4	21.3
RFAG-YOLO [11]	38.9	23.1	5.9	15.7
AeroLight (Ours)	39.8	22.6	1.83	11.5

**Table 3 sensors-25-05369-t003:** VisDrone2019 Validation Ablation Performance. We quantify the contribution of the Pyramid Redesign (P2/P5), the Dynamic Feature Fusion (DFF) module, and the Shape-IoU loss.

Configuration	mAP50 (%)	Δ mAP50	mAP50–95 (%)	Params (M)
(A) Baseline (YOLOv12n)	32.3	-	19.3	2.57
(B) Baseline + Pyramid Redesign	37.9	+5.6	22.0	1.83
(C) Baseline + DFF Module	34.2	+1.9	20.1	2.65
(D) Baseline + Shape-IoU	33.9	+1.6	19.6	2.57
(E) AeroLight (B+C+D)	39.8	+7.5	22.6	1.83

**Table 4 sensors-25-05369-t004:** Performance on the RSOD dataset.

Method	mAP50 (%)	Precision (%)	Recall (%)
RetinaNet	91.7	92.3	90.2
YOLO11n	92.7	93.9	91.2
YOLOv12n	94.5	93.9	91.2
AeroLight (Ours)	95.6	94.2	91.9

**Table 5 sensors-25-05369-t005:** Performance on the Huaxing Farm Drone Dataset.

Method	mAP50 (%)	Precision (%)	Recall (%)
YOLOv12n	80.6	79.8	80.4
AeroLight (Ours)	83.7	83.2	84.1

## Data Availability

The data pertinent to this research are available from the corresponding authors upon request. These data are not publicly accessible as they are derived from lab results.

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
