# Peer review of "AeroLight: A Lightweight Architecture with Dynamic Feature Fusion for High-Fidelity Small-Target Detection in Aerial Imagery"

_sensors, 2025, doi:10.3390/s25175369_

Round 1
Reviewer 1 Report
Comments and Suggestions for Authors
This paper proposes a lightweight architecture specifically designed for small-object detection in UAV aerial imagery. The method is technically interesting, but the manuscript requires substantial revisions to address critical issues related to experimental transparency, benchmark completeness, contextual consistency with UAV applications, and methodological scope. The detailed comments are as follows:
1. The introduction and related work sections omit many recent advances in UAV-specific YOLO optimizations, particularly methods proposed during 2024–2025. These approaches are directly relevant to AeroLight's contributions, and the authors should update their literature review to reflect the current state of the field and better justify their design choices. Redundant discussions of two-stage detectors can be trimmed to give more space to recent advancements in lightweight one-stage frameworks.
2. In Figure 2, modules such as N1–N5 and P2–P5 are presented as simple colored blocks without annotations. It is recommended that the authors add parameter annotations (e.g., resolution, stride) to improve interpretability.
3. The paper uses a fixed input resolution of 640×640; However, datasets such as VisDrone2019 and RSOD contain images with varying aspect ratios and sizes. The manuscript does not clarify how images are preprocessed (resizing, padding, warping, cropping?). These image adjustment and label transformation steps must be clearly explained for both training and inference, as they directly affect reproducibility.
4. In section 4.1, the authors mention a self-collected UAV dataset but do not show any examples. Similar to Figure 4, a visual example from the custom dataset should be included. In addition, more details should be provided about the dataset itself, including annotated categories and image resolution ect.
5. Although models such as YOLO-Drone, DSAA-YOLO, and GCL-YOLO are mentioned in the literature review, none are included in the quantitative comparison. Comparing only with standard YOLO variants (YOLOv5s, YOLOv8n, YOLOv12n) overlooks many context-specific baselines tailored for UAV-based small object detection. Recent state-of-the-art models are underrepresented in the benchmark.
6. In Section 5.3, the qualitative analysis includes only YOLOv12n and AeroLight. While this shows the effect of each architectural modification, it does not support broader claims, such as improved attention quality or background suppression. Broader cross-method comparisons with other lightweight detectors are needed.
7. The manuscript lacks a standalone discussion section. There is no comparative analysis with related SOTA models in terms of design principles or performance trade-offs, nor is there a discussion of potential limitations. In recent UAV detection research, infrared or multispectral sensors are often used. The current architecture assumes RGB-only input, which should be acknowledged and discussed as a limitation.
The method design is commendable. However, the outdated literature review, narrow benchmark selection, and missing preprocessing details are critical issues that need improvement. Given the issues raised above, I believe the manuscript is not yet ready for publication.
Author Response
- Summary
Thank you very much for taking the time to review this manuscript. Please find the detailed responses below and the corresponding revisions in the re-submitted files.
- Point-by-point response to Comments and Suggestions for Authors
Comments 1: The introduction and related work sections omit many recent advances in UAV-specific YOLO optimizations, particularly methods proposed during 2024–2025. These approaches are directly relevant to AeroLight's contributions, and the authors should update their literature review to reflect the current state of the field and better justify their design choices. Redundant discussions of two-stage detectors can be trimmed to give more space to recent advancements in lightweight one-stage frameworks.
Response 1: Thank you for pointing this out. We agree with this comment. Therefore, we have updated the literature on the latest advancements in this field in the Introduction and Related Work sections. This change can be found on pages 1-4 of the revised manuscript. Additionally, we have removed redundant discussions on two-stage detectors and added the latest progress on lightweight single-stage frameworks, which have been highlighted in red and can be found on page 3 of the revised manuscript.
Comments 2: In Figure 2, modules such as N1–N5 and P2–P5 are presented as simple colored blocks without annotations. It is recommended that the authors add parameter annotations (e.g., resolution, stride) to improve interpretability.
Response 2: Thank you for pointing this out. We agree with this comment. Therefore, we have reconstructed Figure 2 and added necessary parameter annotations to make the figure explanatory. This change has been highlighted in red and can be found on page 6 of the revised manuscript.
Comments 3: The paper uses a fixed input resolution of 640×640; However, datasets such as VisDrone2019 and RSOD contain images with varying aspect ratios and sizes. The manuscript does not clarify how images are preprocessed (resizing, padding, warping, cropping?). These image adjustment and label transformation steps must be clearly explained for both training and inference, as they directly affect reproducibility.
Response 3: Thank you for pointing this out. We agree with this comment. Therefore, we have added necessary descriptions of dataset preprocessing to enhance the reproducibility of the experiments. This change has been highlighted in red and can be found on page 9 of the revised manuscript.
Comments 4: In section 4.1, the authors mention a self-collected UAV dataset but do not show any examples. Similar to Figure 4, a visual example from the custom dataset should be included. In addition, more details should be provided about the dataset itself, including annotated categories and image resolution ect.
Response 4: Thank you for pointing this out. We agree with this comment. Therefore, we have provided a brand-new introduction and presentation of the our UAV dataset, and revised it to the professional name: Huaxing Farm Drone Dataset. This change has been highlighted in red and can be found on page 8 of the revised manuscript.
Comments 5: Although models such as YOLO-Drone, DSAA-YOLO, and GCL-YOLO are mentioned in the literature review, none are included in the quantitative comparison. Comparing only with standard YOLO variants (YOLOv5s, YOLOv8n, YOLOv12n) overlooks many context-specific baselines tailored for UAV-based small object detection. Recent state-of-the-art models are underrepresented in the benchmark.
Response 5: Thank you for pointing this out. We agree with this comment. Therefore, we have supplemented the comparative experiments, added specific comparisons of UAV aerial photography algorithms, as well as comparisons between the latest state-of-the-art models and corresponding horizontal models. This change has been highlighted in red and can be found on page 10 of the revised manuscript.
Comments 6: In Section 5.3, the qualitative analysis includes only YOLOv12n and AeroLight. While this shows the effect of each architectural modification, it does not support broader claims, such as improved attention quality or background suppression. Broader cross-method comparisons with other lightweight detectors are needed.
Response 6: Thank you for pointing this out. We agree with this comment. Therefore, we have made significant revisions and improvements to Section 5.3, supplemented multiple sets of comparative experiments, especially improved Figures 6 and 7, and added other lightweight detectors for more extensive cross-method comparisons. This change has been highlighted in red and can be found on page 12 of the revised manuscript.
Comments 7: The manuscript lacks a standalone discussion section. There is no comparative analysis with related SOTA models in terms of design principles or performance trade-offs, nor is there a discussion of potential limitations. In recent UAV detection research, infrared or multispectral sensors are often used. The current architecture assumes RGB-only input, which should be acknowledged and discussed as a limitation.
Response 7: Thank you for pointing this out. We agree with this comment. Therefore, we have added a new discussion section, which elaborates on the practical effects of the algorithm improvements, their limitations, and future improvement plans. This change has been highlighted in red and can be found on page 15 of the revised manuscript.
- Response to Comments on the Quality of English Language
Response : We have seriously edit this paper to guarantee the quality of English language. Thanks again.
- Additional clarifications
Response : We have carefully checked the content and grammar of the entire manuscript, cited the latest references, and revised the descriptions of dataset preprocessing. Meanwhile, by supplementing comparative experiments and their corresponding results as well as adding a new discussion section, we have further improved the structure of the article. Thanks again.

Reviewer 2 Report
Comments and Suggestions for Authors
This paper proposes AeroLight, a lightweight UAV object detection framework that combines a high-resolution feature pyramid redesign, a dynamic feature fusion module, and a shape-aware loss to significantly enhance small-target detection accuracy while reducing computational cost. The paper is organized and easy to follow. However, there are still many deficiencies that require a major revision.
(1) The description of the self-collected high-altitude dataset is too brief, lacking details such as resolution, imaging sensor specifications, weather conditions, etc., which affects reproduction and result interpretation.
(2) It is suggested to add horizontal comparisons of more lightweight models in the experiment to demonstrate that your design superiority is not limited to the YOLOv12n baseline.
(3) The visualization results in Figures 6 and 7 are relatively unclear, and it is recommended to zoom in locally to demonstrate the superiority of the method.
(4) The latest lightweight detection and feature fusion detection methods should be considered in the related work or performance comparison experiment part. The following articles are recommended to be referred:
â‘ CM-YOLO: Typical Object Detection Method in Remote Sensing Cloud and Mist Scene Images (doi. org/10.3390/rs17010125).
â‘¡ Complementarity-Aware Feature Fusion for Aircraft Detection via Unpaired Opt2SAR Image Translation (doi: 10.1109/TGRS.2025.3578876).
(5) Attention should be paid to the standardization of expression, such as reducing font bolding in the main text and adding citations to necessary literature such as “MobileNet, ShuffleNet, and GhostNet”.
Author Response
- Summary
Thank you very much for taking the time to review this manuscript. Please find the detailed responses below and the corresponding revisions in the re-submitted files.
- Point-by-point response to Comments and Suggestions for Authors
Comments 1: The description of the self-collected high-altitude dataset is too brief, lacking details such as resolution, imaging sensor specifications, weather conditions, etc., which affects reproduction and result interpretation.
Response 1: Thank you for pointing this out. We agree with this comment. Therefore, we have provided a brand-new introduction and presentation of the our high-altitude dataset, and revised it to the professional name: Huaxing Farm Drone Dataset. This change has been highlighted in red and can be found on page 8 of the revised manuscript.
Comments 2: It is suggested to add horizontal comparisons of more lightweight models in the experiment to demonstrate that your design superiority is not limited to the YOLOv12n baseline.
Response 2: Thank you for pointing this out. We agree with this comment. Therefore, we have supplemented the comparative experiments, added specific comparisons of UAV aerial photography algorithms, as well as more horizontal comparisons with lightweight models to demonstrate that the design advantages are not limited to the YOLOv12n baseline. This change has been highlighted in red and can be found on page 10 of the revised manuscript.
Comments 3: The visualization results in Figures 6 and 7 are relatively unclear, and it is recommended to zoom in locally to demonstrate the superiority of the method.
Response 3: Thank you for pointing this out. We agree with this comment. Therefore, we have partially enlarged the visualization results of Figures 6 and 7 to reflect the superiority of this method, and added supplements to the experimental results of the corresponding horizontal models, making the results of the article more complete. This change has been highlighted in red and can be found on page 12 of the revised manuscript.
Comments 4: The latest lightweight detection and feature fusion detection methods should be considered in the related work or performance comparison experiment part. The following articles are recommended to be referred:
â‘ CM-YOLO: Typical Object Detection Method in Remote Sensing Cloud and Mist Scene Images (doi. org/10.3390/rs17010125).
â‘¡ Complementarity-Aware Feature Fusion for Aircraft Detection via Unpaired Opt2SAR Image Translation (doi: 10.1109/TGRS.2025.3578876).
Response 4: Thank you for pointing this out. We agree with this comment. Therefore, we have carefully studied and researched the papers provided by the reviewers in the Related Work and Results and Analysis, made revisions in accordance with the provided literature, and cited the corresponding parts. (For example, the structure of visualization result construction, the writing method of literature citation, etc.)
Comments 5: Attention should be paid to the standardization of expression, such as reducing font bolding in the main text and adding citations to necessary literature such as “MobileNet, ShuffleNet, and GhostNet”.
Response 5: Thank you for pointing this out. We agree with this comment. Therefore, we have improved the standardization of the article's expression, reduced the use of bold fonts in the main text, and increased the citations of necessary literature.
- Additional clarifications
Response : We have carefully checked the content and grammar of the entire manuscript, cited the latest references, and revised the descriptions of dataset preprocessing. Meanwhile, by supplementing comparative experiments and their corresponding results as well as adding a new discussion section, we have further improved the structure of the article. Thanks again.

Reviewer 3 Report
Comments and Suggestions for Authors
Revision is necessary.
Author Response
- Summary
Thank you very much for taking the time to review this manuscript. Please find the detailed responses below and the corresponding revisions in the re-submitted files.
- Point-by-point response to Comments and Suggestions for Authors
Comments : Revision is necessary.
Response : Thank you for pointing this out. We agree with this comment. Therefore, we have carefully checked the content and grammar of the entire manuscript. On this basis, we have further refined and optimized the research design, comprehensively upgraded and improved the quality of figures and tables in the article, supplemented the latest references, and optimized the descriptions related to dataset preprocessing. Meanwhile, by adding comparative experiments and their corresponding results, as well as including a new discussion section, we have made the structure of the article more rigorous and complete.All these changes have been highlighted in red throughout the revised manuscript.
- Additional clarifications
Response : We have significantly optimized the visualization results of the model, with a particular focus on enhancing the presentation of Figures 6, 7, and 11. Meanwhile, we have refined the description of the dataset to be more rigorous and detailed.

Round 2
Reviewer 1 Report
Comments and Suggestions for Authors
All my comments have been addressed during this round of revisions, and I think the manuscript is now ready for publication.
Author Response
- Summary
Thank you very much for taking the time to review this manuscript. Please find the detailed responses below and the corresponding revisions in the re-submitted files.
- Point-by-point response to Comments and Suggestions for Authors
Comments: All my comments have been addressed during this round of revisions, and I think the manuscript is now ready for publication.
Response: Thank you for your valuable comments and suggestions! These insightful and helpful insights have not only significantly enhanced the completeness of the article but also made the details more delicate and precise. Please accept my sincere gratitude.

Reviewer 2 Report
Comments and Suggestions for Authors
(1) Some methods in the performance comparison section are not labeled with necessary references, such as TOOD, GFL, etc.
(2) The abstract section is suggested to remove bold fonts to optimize the standardization of expression.
(3) It is suggested to optimize Figure 1 to highlight modules related to innovation points.
(4) In the part of feature fusion and attention mechanism work, it is recommended to consider more feature fusion detection methods released in 2025 to demonstrate the adequacy of method research.
â‘ Complementarity-Aware Feature Fusion for Aircraft Detection via Unpaired Opt2SAR Image Translation (doi: 10.1109/TGRS.2025.3578876).
â‘¡Towards robust infrared small target detection via frequency and spatial feature fusion (doi: 10.1109/TGRS.2025.3535096).
Author Response
- Summary
Thank you very much for taking the time to review this manuscript. Please find the detailed responses below and the corresponding revisions in the re-submitted files.
- Point-by-point response to Comments and Suggestions for Authors
Comments 1: Some methods in the performance comparison section are not labeled with necessary references, such as TOOD, GFL, etc.
Response 1: Thank you for pointing this out. We agree with this comment. Therefore, we have added necessary references for some methods in the performance comparison section. This change has been highlighted in red and can be found on page 10 of the revised manuscript.
Comments 2: The abstract section is suggested to remove bold fonts to optimize the standardization of expression.
Response 2: Thank you for pointing this out. We agree with this comment. Therefore, we have removed the bold fonts in the abstract and optimized the standardization of expressions.
Comments 3: It is suggested to optimize Figure 1 to highlight modules related to innovation points.
Response 3: Thank you for pointing this out. We agree with this comment. Therefore, we have optimized Figure 1, highlighted the relevant innovative modules with red boxes, and added explanations in the legend. This change has been highlighted in red and can be found on page 5 of the revised manuscript.
Comments 4: In the part of feature fusion and attention mechanism work, it is recommended to consider more feature fusion detection methods released in 2025 to demonstrate the adequacy of method research.
â‘ Complementarity-Aware Feature Fusion for Aircraft Detection via Unpaired Opt2SAR Image Translation (doi: 10.1109/TGRS.2025.3578876).
â‘¡Towards robust infrared small target detection via frequency and spatial feature fusion (doi: 10.1109/TGRS.2025.3535096).
Response 4: Thank you for pointing this out. We agree with this comment. Therefore, in the sections on feature fusion and attention mechanism, we have referred to the feature fusion detection method published in 2025 and cited the paper you provided to demonstrate the sufficiency of the method research. This change has been highlighted in red and can be found on page 4 of the revised manuscript.
- Additional clarifications
Response : Thank you for your valuable comments and suggestions! We have made careful revisions to the corresponding sections. These insightful and highly valuable comments have not only significantly enhanced the completeness of the article but also made the presentation of details more accurate and refined. We would like to express our most sincere gratitude to you once again.
